# Limiting spread of COVID-19 in Ghana: Compliance audit of selected transportation stations in the Greater Accra region of Ghana

Harriet Affran Bonful[1], Adolphina Addo-Lartey[1], Justice M. K. Aheto[2], John Kuumouri Ganle[3], Bismark Sarfo[1], Richmond Aryeetey[3]*

1 Department of Epidemiology and Disease Control, University of Ghana School of Public Health, Accra, Ghana, 2 Department of Biostatistics, University of Ghana School of Public Health, Accra, Ghana, 3 Department of Population and Family Health, University of Ghana School of Public Health, Accra, Ghana

* raryeetey@ug.edu.gh

## Abstract

Globally, little evidence exists on transmission patterns of COVID-19. Recommendations to prevent infection include appropriate and frequent handwashing plus physical and social distancing. We conducted an exploratory observational study to assess compliance with these recommendations in selected transportation stations in Ghana. A one-hour audit of 45 public transport stations in the Greater Accra region was carried out between 27th and 29th March 2020. Using an adapted World Health Organization (WHO) hand hygiene assessment scale, the availability and use of handwashing facilities, social distancing, and ongoing public education on COVID-19 prevention measures were assessed, weighted and scored to determine the level of compliance of stations. Compliance with recommendations was categorized as "inadequate" "basic", "intermediate" and "advanced", based on the overall score. Majority (80%) of stations in Accra have at least one Veronica Bucket with flowing water and soap, but the number of washing places at each station is not adequate. Only a small minority (18%) of stations were communicating the need to wash hands frequently and appropriately, and to practice social/physical distancing while at the station. In most stations (95%), hand washing practice was either not observed, or only infrequently. Almost all stations (93%) did not have alcohol-based hand sanitizers available for public use, while social distancing was rarely practiced (only 2%). In over 90% of the stations, face masks were either not worn or only worn by a few passengers. Compliance with COVID-19 prevention measures was inadequate in 13 stations, basic in 16 stations, intermediate in 7 stations, and advanced in 9 stations. Compliance with COVID-19 prevention measures in public transportation stations in the Greater Accra region remains a challenge. Awareness creation should aim to elevate COVID-19 risk perception of transportation operators and clients. Transport operators and stations need support and guidance to enforce hand washing and social distancing.

**Data Availability Statement:** The dataset has been made publicly available at Figshare and the digital

object identifier (DOI) is https://doi.org/10.6084/m9.figshare.12761984.

**Funding:** Funding for the study was obtained from the University of Ghana Office of Research, Innovation, and Development. The Initial of recipient is RA. The URL of the funder site is http://orid.ug.edu.gh/. The funders had no role in study design, data collection and analysis, decision to publish, or preparation of the manuscript.

**Competing interests:** The authors have declared that no competing interests exist.

**Abbreviations:** FDA, Food and Drugs Authority; GAR, Greater Accra Region; GHS, Ghana Health Service; MOH, Ministry of Health; PPE, Personal Protective Equipment; WHO, World Health Organization.

## Introduction

Coronavirus disease 2019 (COVID-19) is an infectious disease caused by coronaviruses, specifically, severe acute respiratory syndrome coronavirus 2 (SARS-CoV-2) [1,2]. From the time when the disease was first reported in the Wuhan Province in China in December 2019 [2], it has affected more than eighteen million people globally, with over six hundred thousand deaths [3]. The World Health Organization (WHO) declared COVID-19 as a pandemic on 11th March 2020 [4]. COVID-19 is a highly transmissible disease with a basic reproductive number estimated to be higher than that of Severe Acute Respiratory Syndrome (SARS), which only affected 26 countries and caused about 8,000 deaths in 2002 [5,6].

COVID-19 is transmitted from person to person through small droplets from the nose or mouth, which are expelled when a person with COVID-19 coughs, sneezes, or speaks and also via contact with fomites [2,7]. The virus has been shown to survive outside a host for durations that depend on the nature of the surface. It can survive in the air for up to 3 hours, on copper surfaces for up to 4 hours, on cardboard for up to 24 hours, and plastic and stainless steel, for up to 72 hours [8]. Common symptoms of COVID-19 include fever, cough, colds, headaches, and difficulty in breathing. Available evidence suggests that the pathogenicity of SARS-COV2 depends on host factors such as age and other comorbidities [9–12]. There is, currently, no approved treatment for COVID-19. Neither is there any vaccine for prevention in vulnerable populations [2].

The first two cases of COVID-19 in Ghana were identified on the 12th of March 2020 [13]. By April 19th, more than 1,000 confirmed cases of COVID-19 and nine deaths had been reported. To reduce person-to-person transmission, the Government of Ghana adopted and promoted the WHO's recommendations [14], which include avoiding or limiting physical contact (including handshake and other forms of usual contact), regular handwashing with soap under running water, rubbing of hands with alcohol-based sanitizers with 70% alcohol strength, and reducing/limiting large gatherings among the general populace. Coughing into the elbow or tissue and disposing it immediately into a bin have also been recommended. Preventive behavioral change messages have been developed and are being disseminated through various media (radio, television, social media, and print media), nation-wide.

Emphasis has been placed on ensuring adequate handwashing and social distancing in all public places, including markets and transport terminals. This was partly because the majority of urban-dwelling Ghanaians rely on open markets for groceries and informal public transportation for daily commuting. Public transportation stations in many parts of the Greater Accra Region (GAR) including Accra and Tema are usually not spacious and are characterized by high vehicular and human density, especially during rush- hours. Also, they are mostly owned and managed by private individuals, resulting in little or no risk management by city authorities.

The public transport system is essentially informal and privately-managed by independent operator unions, and designed to convey intra-city commuters using Mini-buses and Taxis. The city is also served by large capacity buses for travel between cities. Irrespective of the category of transportation, passengers often need to converge at crowded stations to access transportation. During rush hours (6-9am in the morning and 4-7pm in the evening), many commuters congregate at stations, and often have to wait in queues to access public transportation to various destinations in Accra, Tema, and other administrative capitals in the region. This arrangement creates large crowded situations that limit the ability to effectively practice social distancing. While onboard the vehicles, passengers usually sit or stand very close to each other, largely, because of overloading. This situation further creates a fertile environment for spread of COVID-19 transmission.

With recognition of the government's recommendations to limit the spread of COVID-19, it is critical to assess public responses to these preventive measures. At the time of our study, compliance with these preventive measures, especially in urban spaces where intense human interaction takes place, had not been systematically evaluated. We, therefore, assessed ecological readiness and compliance to hand washing, and social and physical distancing recommendations in selected public transportation stations in the GAR. Such a study is urgently needed to provide evidence to guide policy and behavior change communication aimed at reducing the spread of COVID-19 in Ghana and similar settings.

## Materials and methods

### Design and sampling

The study was a descriptive observational compliance audit of the level of preparedness and compliance with hygiene and social distancing recommendations in public transport stations. The GAR has a population of almost five million with over 137 registered market centers and their corresponding public transportation stations [15,16]. There are 2 metropolises, 9 municipalities and 5 districts, making a total of 16 administrative units in the GAR. These administrative units differ significantly in size, and volume of business activities, which is usually highest at the metropolitan level and lowest in a district.

To ensure that our sample reflects this heterogeneity in the GAR, of the 16 administrative units in the GAR, 11 were included in the study. They included two metropolitan cities (Accra and Tema), seven municipal cities (Ashaiman, Kpone-Katamanso, La Nkwantanang Madina, Ayawaso West, Ablekuma North, Ablekuma South, and Ga East) and two districts (Ningo-Prampram, and Ga West).

There is no existing documented evidence of daily passenger traffic across stations in the country. Within each of the selected units, based on the judgement of the authors, the lorry stations that are generally noted to carry a relatively high daily passenger traffic in the intra- city transportation sector were selected.

Observations were carried in a total of 45 commercial transport stations. These stations were purposively selected based on their size and volume of daily passenger traffic. The names of the lorry stations are listed in Fig 1.

### Data collection

The data collection tool was developed by adapting questions from the WHO Hand Hygiene Self-Assessment Framework [17]. The revised tool had a total of 26 question items, distributed across six sections. The first section assessed the communication of hand hygiene and social distancing at transport stations. Four observational question items were used, with a minimum score of 0 and a maximum score of 15 per question, giving a maximum possible score of 60. The second section assessed the availability of handwashing facilities. Four observational question items were used here, with a minimum section score of 0 and a maximum possible score of 45. The third section assessed the availability of water for handwashing based on three observational question items. The minimum section score was 0 and the maximum possible score was 30. The fourth section assessed the availability of soap, and hand sanitizers. Two observational question items were used, with a minimum section score of 0 and a maximum possible score of 20. The fifth section assessed the utilization of handwashing amenities using four-question items. The minimum section score was 0 and the maximum possible section score was 35. The final section assessed social distancing based on nine observational question items, with a minimum section score of 0 and a maximum possible score of 55.

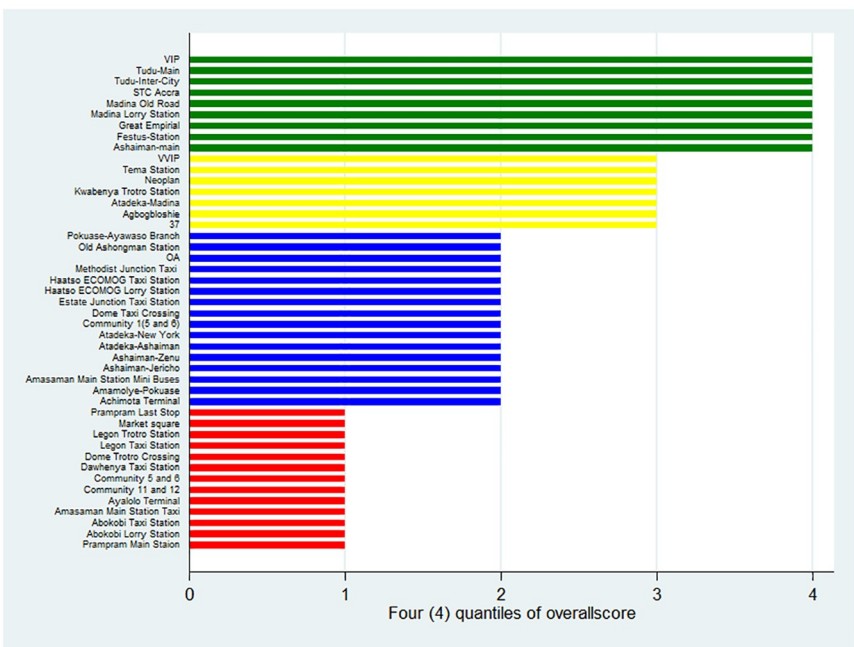

**Fig 1. Compliance of transport stations distributed by quantiles (n = 45).**

The tool was pretested at one commercial transport station in Accra (Atomic Roundabout Station), and revised using the findings and feedback from the pre-test. Data were collected between 27th and 29th March 2020, just before an anticipated public mobility restriction executive order, came into force on 30th March 2020. Four of the authors of this manuscript collected the data. To ensure standardization in the data collection processes, the research team visited stations at three specific time points: 8:00am-10:00am; 12; 00–2:00pm, and 3:00pm-5:00pm. These periods were chosen to correspond to the peak periods in most public transportation stations in urban Ghana. Observation and compliance auditing lasted at least one hour at each station. Compliance with COVID-19 prevention recommendations was assessed in terms of identifying ongoing public education about hand washing and social distancing at transportation stations, availability of handwashing facilities, water, detergents (soap and sanitizers), use of handwashing facilities, and social distancing.

At each station, the observer walked along all lanes and observed the availability of hygiene facilities, source of water (veronica bucket, running water or other means), cleanliness of the water, number of handwashing facilities, and frequency of handwashing, whether clients washed their hands with water alone or with water and soap, and overcrowding at the handwashing points. Where notice boards were available at the vehicle station, they were checked for posters with messages on COVID-19 prevention as well as proper handwashing procedures and how to wear a nose mask. The research team also listened to determine whether any information is being aired through mobile or stationary public address systems to determine whether public announcements or educational messages on COVID-19 were being disseminated at the stations.

We also assessed social distancing practice among drivers, load bearers, passengers, and vendors at the stations. The availability of posters promoting social distancing, and any infrastructural or spatial changes including barricades and systematically spaced seating

arrangements aimed at ensuring social distancing while passengers waited in queues to board vehicles were checked. Passengers boarding or un-boarding vehicles were observed closely to determine if there were efforts not to touch surfaces that can lead to spreading of the virus e.g. car doors, seats, station chairs. Also, wearing nose masks or other similar Personal Protective Equipment (PPE) was observed.

### Data management and analysis

Questionnaires were manually checked for completeness and entered in excel 2013, where data cleaning, validation, and quality checks were done. The data were then imported into STATA version 14.2 (Stata Corporation, College Station, Texas) for further management and analyses. Internal consistency checks were first conducted to ensure the validity and completeness of the data before analysis. Descriptive statistics were used to summarize the availability of hygiene facilities, water, detergents, and use of the available handwashing facilities and observation of social distancing. To further understand the level of preparedness and compliance with different aspects of recommended COVID-19 prevention measures at the transportation stations, we calculated the overall total score and total section score for each of the six different components/sections of our assessment. The overall potential total score (240) was converted into quantiles of four (4), with 1 (one) representing the lowest or first quantile and 4 (four) representing the highest or the fourth quantile.

Compliance was deemed 'Inadequate' 'Basic', 'Intermediate" or 'Advanced' if the overall total score fell within the 1st, 2nd, 3rd, or 4th quantile respectively. Proportions were then used to describe compliance of stations.

## Results

### Characteristics of transportation stations assessed

A total of 45 transportation stations were assessed. Table 1 shows the characteristics of these stations and their location. Nearly half of the stations (47%) were mini-bus stations. The Accra Metropolitan Assembly had the highest number of public transportation stations included among those observed (22.2%).

### Communication on observing personal hygiene

Table 2 shows that most of the stations (82%) had not provided any printed communication, (ie notices/posters) with information on appropriate hand hygiene practice. Although most stations use audio systems to manage their operations, audio announcements about handwashing/personal hygiene were made in only one (2.5%) station (Tudu Inter-city) during the observation period.

### Availability of hand hygiene amenities at transportation stations

Table 3 shows data on the availability of hand hygiene facilities at the 45 public transportation stations that were studied. Most of the stations (84%) had installed a handwashing facility at the time of observation. Among the 38 stations that had installed a handwashing facility, the majority (53%, n = 20) had only one spot for hand washing. Most of the installed handwashing facilities (90%, n = 34) were Veronica Buckets with receptacles for collecting wastewater. Running water and soap (solid/liquid) were available in many of the stations with installed handwashing facilities (93% and 90%, respectively).

**Table 1. Characteristics of public transportation stations (n = 45).**

| Characteristics | Frequency | Percent |
|---|---|---|
| **Station Type** | | |
| Taxi | 17 | 37.7 |
| Mini buses | 21 | 46.7 |
| Long buses | 7 | 15.6 |
| Total | 45 | 100 |
| **District/Municipality** | | |
| Accra Metropolitan Assembly | 10 | 22.2 |
| Ashiama Municipal | 4 | 8.9 |
| Ayawaso West Municipal | 3 | 6.7 |
| Ablekuma North Municipal | 1 | 2.2 |
| Ablekuma South Municipal | 1 | 2.2 |
| Ga West District (Amasaman) | 5 | 11.1 |
| Ga East Municipal | 8 | 17.8 |
| KponeKatamanso District | 3 | 6.7 |
| La NkwantanangMadina Municipal | 2 | 4.4 |
| Ningo-Prapram District | 5 | 11.1 |
| Tema Metropolis | 3 | 6.7 |
| **Total** | **45** | **100.0** |

## Utilization of handwashing facilities and sanitizers

As shown in Table 4, in the 38 stations where handwashing facilities were available, there was no observation of the facilities being used in 5% of the stations (n = 2). In the stations where they were used at least once, almost all the facilities were used rarely 87% (n = 34). Only in 5% (n = 2) of the stations were the handwashing facilities used frequently. Soap was available in 34

**Table 2. Hand hygiene communication at selected transport stations in the Greater Accra Region (n = 45).**

| Observation item | Frequency | Percent |
|---|---|---|
| **Posters with information on hand hygiene** | | |
| Not displayed at all | 37 | 82.2 |
| Displayed in some areas | 6 | 13.3 |
| Displayed in most areas | 2 | 4.4 |
| **Posters explaining correct hand washing techniques** | | |
| Not displayed at all | 41 | 91.1 |
| Displayed in some areas | 3 | 6.7 |
| Displayed in most areas | 1 | 2.2 |
| **Other hygiene reminders (e.g. coughing or sneezing into tissue paper/elbow)** | | |
| Not displayed at all | 38 | 84.4 |
| Displayed in some areas | 4 | 8.9 |
| Displayed in most areas | 3 | 6.7 |
| **Audio announcements about handwashing /personal hygiene** | | |
| No announcement at all | 44 | 97.8 |
| Announcement made only once | 0 | 0.0 |
| Announcement made severally | 1 | 2.2 |
| **Total** | **45** | **100.00** |

**Table 3. Availability of hand hygiene amenities at transport stations in the Greater Accra Region.**

| Observation item | Frequency | Percent |
|---|---|---|
| **At least one installed handwashing facility (n = 45)** | 38 | 84.4 |
| **Number of places for handwashing (n = 38)** | | |
| Only one for the entire station | 20 | 52.6 |
| More than one for the entire station | 18 | 47.4 |
| **Nature of handwashing place (n = 38)** | | |
| Ceramic Sink with a tap | 1 | 2.6 |
| Veronica bucket with receptacle only | 34 | 89.5 |
| Sink and Veronica bucket at the same station | 2 | 5.3 |
| Others[1] | 1 | 2.6 |
| **Hand washing facility is accessible at station (n = 38)** | 37 | 88.1 |
| **Running water available for handwashing place (n = 38)** | 35 | 92.7 |
| **Available water is visibly clean (n = 35)** | 35 | 100.0 |
| **Soap(solid/liquid) is available for handwashing (n = 38)** | 34 | 90.0 |
| **Availability of alcohol-based hand sanitizer (s) (n = 44)** | | |
| None | 41 | 93.1 |
| Available at one location in the station | 2 | 4.6 |
| Available at more than one location in the station | 1 | 2.3 |

[1](Polytank/ Large rubber gallons with water).

stations but they were used in only 87% (n = 30) of the stations observed. Use of alcohol-based hand sanitizer when boarding/un-boarding vehicles was observed at only three stations (7%). We did not observe the availability and use of other types of sanitizers, apart from alcohol-based sanitizers which had been recommended by the government.

## Social distancing at transportation stations

As indicated in Table 5, two stations (5%) provided communication with messages promoting social distancing. Only one station (State Transport Corporation, Accra) had infrastructural re-arrangements to enable social distancing. It was only in two stations that we observed passengers actively exercising physical distancing from each other at the station. In the majority of stations (63%, n = 27), only a few passengers were observed wearing personal protective equipment. We observed the use of handkerchiefs, headgears, and personal clothing being used as face masks.

**Table 4. Utilisation of handwashing facilities at public transportation stations in the Greater Accra Region.**

| Observation item | Frequency | Percent |
|---|---|---|
| **Use of handwashing facilities (n = 38)** | | |
| Not used | 2 | 5.3 |
| Infrequently used | 34 | 87.4 |
| Frequently used | 2 | 5.3 |
| **Used soap when washing hands (n = 36)** | 30 | 83.3 |
| **Use of alcohol-based hand sanitizer when boarding/un-boarding buses/cars (n = 44)** | 3 | 6.8 |

**Table 5. Social distancing at public transportation stations in the Greater Accra Region of Ghana.**

| Observation item | Frequency | Percent |
|---|---|---|
| **Visible/recognizable communication/messages on social distancing at station** (*e.g. poster/audio message*) (**n = 44**) | 2 | 4.6 |
| **Infrastructural or spatial changes to ensure social distancing at the station** (*e.g. barricades for how to stand in queues*) (**n = 45**) | 1 | 2.2 |
| **Arrangements by Public transportation operators to promote social distancing** (*e.g. enforcing appropriate queuing, boarding or seating arrangements*) (**n = 45**) | 1 | 2.2 |
| **Passengers maintaining social distance from other passengers within transportation stations** (e.*g. deliberate individual attempts to maintain a reasonable distance from other people*) (**n = 43**) | 1 | 2.3 |
| **Other persons in transportation stations (including vendors, load bearers) observing social distance when interacting with passengers(n = 44)** | 3 | 93.2 |
| **Wearing of protective clothing/equipment (PPEs) within transportation station** (*e.g. nose mask or other similar PPEs*)(**n = 43**) | | |
| Not worn at all | 15 | 34.9 |
| Worn by a few | 27 | 62.8 |
| Worn by many | 1 | 2.3 |
| **Passengers making effort not to touch surfaces (e.g. car doors, seats, station chairs) (n = 43)** | 1 | 2.3 |
| **Passengers were seen making an effort to keep a social distance from vendors in the station (n = 42)** | 2 | 4.8 |

## Overall compliance of transportation stations to COVID-19 preventive measures

Fig 1 shows that the top 10 performing public transportation stations based on the overall scores for handwashing and social distancing recommendations performance were Ashaiman-Main Station, Festus-Station, Great Imperial, Madina Zongo Junction Lorry Station, Madina Old Road, STC Accra, Tudu-Inter-City, Tudu-Main, and VIP—Circle. The worst complaint stations were Prampram Main Station, Abokobi Lorry Station, Abokobi Taxi Station, Amasaman Main Station Taxi, Ayalolo Terminal, Community 11 and 12, Community 5 and 6, Dawhenya Taxi Station, Dome Trotro Crossing, Legon Taxi Station, Legon Trotro Station, Market Square, and Prampram Last Stop.

In terms of the total thematic compliance score for all the 45 stations, Table 6 shows that majority (82%) of the stations were classified as belonging to the first quantile regarding the personal hygiene communication. Eight stations (18%) were classified as belonging to the fourth quantile for this theme. Regarding availability of hygiene facilities, 64% (29), 31% (14), and 4% (2) were classified in the first, third, and fourth quantiles respectively. Almost all (96%) of stations belonged to the first quantile group with only 2 stations (4%) belonging to the fourth quantile group for the availability of detergents at the stations. Regarding hand washing facilities and hand sanitizer, 29% (13), 67% (30), and 4% (2) of the stations were identified with the first, second, and fourth quantiles respectively. Many (56%) of the stations were classified as belonging to the second quantile while 33% (25) were classified as belonging to the first quantile for social distancing. Also, 11% (5) were classified as belonging to the fourth quantile for social distancing.

For the overall compliance score, 29%(13), 36%(16), 16%(7), and 20%(9) of the stations were classified as belonging to first, second, third, and fourth quantiles respectively (Table 6). Compliance with COVID-19 prevention measures was classified as inadequate in 13 stations, basic in 16 stations, intermediate in 7 stations, and advanced in 9 stations.

**Table 6. Sectional and overall compliance of transport stations distributed by quantiles (n = 45).**

| Categories | Frequency | Percent |
|---|---|---|
| *Personal hygiene education/announcement* | | |
| Inadequate hygiene communication | 37 | 82.2 |
| Advanced hygiene communication | 8 | 17.9 |
| *Availability of hygiene facilities* | | |
| Inadequate hygiene facilities | 29 | 64.4 |
| Intermediate hygiene facilities | 14 | 31.1 |
| Advanced hygiene facilities | 2 | 4.4 |
| **Availability of detergents** | | |
| Inadequate detergent status | 43 | 95.6 |
| Advanced detergent status | 14 | 4.4 |
| **Use of handwashing facilities & hand sanitizer** | | |
| Inadequate handwashing facilities | 13 | 28.9 |
| Basic handwashing facilities | 30 | 66.7 |
| Advanced handwashing facilities | 2 | 4.44 |
| **Social distancing** | | |
| Inadequate social distancing | 15 | 33.33 |
| Basic social distancing | 25 | 55.56 |
| Advanced social distancing | 5 | 11.11 |
| **Overall score** | | |
| Inadequate station performance | 13 | 28.89 |
| Basic station performance | 16 | 35.56 |
| Intermediate station performance | 7 | 15.56 |
| Advanced station performance | 9 | 20.00 |

## Discussion

Early in the pandemic, Ghana was identified among African countries with the highest vulnerability, as well as limited capacity to respond to the COVID-19 pandemic [18,19] Public transportation is an indispensable service that must continue during a COVID-19 outbreak situation. A key outcome of the study is that majority (80%) of public transportation stations have at least one Veronica Bucket with flowing water and soap. While this effort to ensure handwashing by providing facilities is in line with recommended actions, passengers were not observed actively using these facilities, or were using them infrequently. Our data demonstrate that it is not sufficient to provide handwashing facilities. It is therefore important to generate demand as well as enforce usage of the hand washing facilities at the point of use [14]. Given the adverse consequences of uncontrolled COVID-19 spread, it may be appropriate to go beyond appealing to station users and managers to use the facilities and to use other means, necessary, to enforce basic hand hygiene practices. This may involve using methods similar to the safety practices utilized in the airline industry to prevent terrorism which have become routine public safety standards for airline transportation, post-September 11.

Proper handwashing is essential to preventing COVID-19 transmission [18]. The World Health Organization (WHO) has recommended that regular and thorough cleaning of hands with an alcohol-based hand sanitizer or washing with soap and water kills viruses that may be on your hands [14]. Unavailability of a sufficient number of handwashing locations, and infrequent handwashing at public transport stations is, therefore, an important public health challenge, as it could lead to the rapid spread of COVID-19. Public transport surfaces such as door

handles, seats, and restrainers are constantly touched by passengers and can be a source of transmission. Indeed, the benefits of good practices at stations that are implementing COVID-19 prevention measures could be eroded by non-compliant stations.

Infrequent hand washing at public transportation stations could be explained by several factors. First, it could be linked to inadequate relevant public education about the importance of handwashing to prevent COVID-19 infection. This is more likely, given that awareness creation about the pandemic started rather late, coupled with a general low-risk perception of COVID-19 community spread. Second, it could be linked to the simple but also cultural fact that people may not be used to washing their hands routinely in public, especially at the lorry stations. It will take time for people to acquire the habit of washing hands frequently. This suggests a need for continuous public education using appropriate local mediums and language to ensure that COVID-19 prevention information, advice, and recommendations are easily accessible and understandable to the wider Ghanaian public. Such education interventions are particularly warranted given that only a small minority of stations (18%) in this study were communicating the need to wash hands frequently and appropriately, and to practice social/physical distancing, while at the station. Insufficient communication by transportation operations could be attributed to the fact that the stations and transport operators may not aware of the role they can play in the national efforts to prevent the spread of COVID-19. Further, they are also not experts with capacity for responding to such a health risk in a systematic fashion. They will therefore need to be supported by the city administration in this regarding.

This study also showed that social distancing was rarely practiced; in fact, it was observed in only one station. One explanation for this could be that there is insufficient risk perception of COVID-19 transmission in the general population as a whole. While low-risk perception is not unusual for a novel disease like COVID-19, it is worrisome, because of the potential risk of infection in such crowded spaces. The government's advice is to maintain a physical distance of 1 meter when interacting with others, but there was no communication and education about this recommendation at the stations observed. Neither was there established, any arrangements to enforce it at the stations. This observed shortfall in compliance with the social distancing at the stations is another reminder of the necessity for intensified communication regarding COVID-19 prevention in Ghana.

We also observed that the majority of passengers were not using any PPE. The non-use of face masks in crowded transportation stations, where the practice of social distancing is almost non-existent is a public health concern [20]. At the time of the study, the government had not mandated use of face masks in public spaces. Since then, use of face mask has become mandatory in Ghana and the Food and Drugs Authority (FDA) of Ghana has issued guidelines for the production of appropriate cloth masks [21]. However, due to the additional cost that procuring a face mask may entail, we may continue seeing handkerchiefs, headgears, and personal clothing being used as face masks by a section of the population.

Specific actions that can contribute to addressing the guidelines adherence gaps observed in the current study include enforcing mandatory wearing of face masks including both transport operators and travellers, frequently cleaning surfaces including door knobs, seats, counter tops, and arm rests of vehicles using appropriate sanitizing agents. Enforcing mandatory hand hygiene will only work if the government, at all levels provides financial incentives and technical support for transport operators to install and enforce use of handwashing and hand sanitizing infrastructure. In addition to these, interventions to ensure appropriate physical distancing and implementation of other relevant guidelines when vehicles are in transit are warranted. Presently, use of face masks is mandated by the City of Accra regulations. Specific physical distancing on board public transportation was also established early in the pandemic. The main challenge to implementing these recommended actions will be enforcement. Considering the

observed poor adherence to the existing guidelines observed in the transport stations, further research is needed to understand the drivers of behavior change related to these guidelines.

The findings of our study should be interpreted with certain limitations in mind. The study captured a snapshot of prevailing levels of compliance which can vary substantially with time depending on the coverage and success of interventions. Also, even though we standardized how observations in the stations should be done, individual biases could have still been introduced in the data collection process. It is possible that some passengers may have used their own personal alcohol-based sanitizers before boarding or alighting from vehicles. However, the study was not designed to measure this behavior. Despite these limitations, we believe our findings apply to other public transport stations across the country, making our research relevant for policy directions.

## Conclusion

The audit of transport stations revealed that compliance with COVID-19 prevention measures in public transportation stations in the Greater Accra region remains a challenge. There is currently limited risk communication and practice of handwashing across almost all stations. While the availability of facilities (i.e. veronica buckets, water, and detergents) was relatively better, washing places were still inadequate. Studies are needed to determine standards on how many washing places are needed in public places like transport stations. Social distancing and wearing of PPE's were also poorly observed in almost all the stations. Based on these findings, it is recommended that awareness creation should aim to elevate COVID-19 risk perception among transportation operators and passengers. State and private sector support and guidance should be provided to transport operators and stations to enforce handwashing, wearing of PPEs, and social distancing. Also, the most compliant stations could be used as best practice models, so that lessons and practices from best-performing stations could be used to improve the situation in poorly performing stations.

## Supporting information

**S1 File.**
(DOCX)

## Acknowledgments

We would like to acknowledge all transportation station managers who provided permission for this study to be carried out.

## Author Contributions

**Conceptualization:** Bismark Sarfo, Richmond Aryeetey.

**Data curation:** Justice M. K. Aheto.

**Formal analysis:** Harriet Affran Bonful, Justice M. K. Aheto.

**Funding acquisition:** Richmond Aryeetey.

**Investigation:** Harriet Affran Bonful, Justice M. K. Aheto, John Kuumouri Ganle, Bismark Sarfo.

**Methodology:** Harriet Affran Bonful, Adolphina Addo-Lartey, Justice M. K. Aheto, John Kuumouri Ganle, Bismark Sarfo, Richmond Aryeetey.

**Project administration:** Harriet Affran Bonful, Richmond Aryeetey.

**Supervision:** Harriet Affran Bonful, Richmond Aryeetey.

**Validation:** Harriet Affran Bonful, Richmond Aryeetey.

**Writing – original draft:** Harriet Affran Bonful, Adolphina Addo-Lartey, Justice M. K. Aheto, John Kuumouri Ganle, Bismark Sarfo, Richmond Aryeetey.

**Writing – review & editing:** Harriet Affran Bonful, Adolphina Addo-Lartey, Justice M. K. Aheto, John Kuumouri Ganle, Bismark Sarfo, Richmond Aryeetey.

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
