## [Decision Letter · Decision Letter 0]

23 Jun 2020

PONE-D-20-13058

Limiting Spread of COVID-19 in Ghana: Compliance audit of selected transportation stations in the Greater Accra region of Ghana

PLOS ONE

Dear Dr. Richmond Aryeetey ,

Thank you for submitting your manuscript to PLOS ONE. After careful consideration, we feel that it has merit but does not fully meet PLOS ONE’s publication criteria as it currently stands. Therefore, we invite you to submit a revised version of the manuscript that addresses the points raised during the review process.

We look forward to receiving your revised manuscript.

Kind regards,

Francesco Di Gennaro

Academic Editor

PLOS ONE

Additional Editor Comments:

Dear Authors, we read with great interest your manuscript.

i believe that have potentiality but need major revision

2. Please provide more information of the selection criteria used to choose the 45 transport stations in this study, along with the names of the 45 transport stations.

3. Please remove "2. There was an anticipated lock-down by the Government that made it impractical to either submit an ethics application or carry out the study as planned without going to the field, immediately. considering there will be no human interaction and the methods do not pose any harm to individuals, authors, therefore judged it ethically appropriate to proceed in order not to loose the opportunity for implementing the study. The decision was also strengthen by the the lack of information on when the lock down will be lifted" from the ethics section of your manuscript, as this is not a valid reason in this case to not obtain ethics approval.

Reviewers' comments:

Reviewer's Responses to Questions

**Comments to the Author**

1. Is the manuscript technically sound, and do the data support the conclusions?

Reviewer #1: Yes

Reviewer #2: No

2. Has the statistical analysis been performed appropriately and rigorously? 

Reviewer #1: Yes

Reviewer #2: No

3. Have the authors made all data underlying the findings in their manuscript fully available?

Reviewer #1: No

Reviewer #2: No

4. Is the manuscript presented in an intelligible fashion and written in standard English?

Reviewer #1: Yes

Reviewer #2: No

5. Review Comments to the Author

Reviewer #1: The outbreak of severe acute respiratory syndrome coronavirus 2 (SARS-CoV-2) or COVID-19 in Wuhan, Hubei Province, China and declared a pandemic by the World Health Organization (WHO) on March 11, 2020, has brought the entire world on its knees. As there is currently no specific therapeutics and vaccines available for disease control, the epidemic of COVID-19 is posing a great threat to global public health. In the absence of any scientifically approved control mechanisms, country-specific preventive measures and interventions were therefore deemed significant in minimizing the spread of the infections. This observational study “Limiting Spread of COVID-19 in Ghana: Compliance audit of selected transportation stations in the Greater Accra region of Ghana” aimed to assess compliance to the WHO adopted preventive recommendations by the Ghana government in selected public transportation stations in Ghana. The study rationale is very sound and appropriate considering the high human congestion observed in the Ghanaian public transport stations. The major findings are that 1) there is non-compliance to COVID-19 preventive measures in these stations, 2) communication deficit, and non-adherence to handwashing, non-observance of social distancing and limited use of nose masks at these stations. While the background was succinctly written, the authors, however, failed to outline the needed stringent preventive measures that transportation managers/owners could adopt at the stations to enhance compliance in the discussion. Another concern is that no ethical approval was sought for this study which is clearly against the Helsinki Declaration of 1975, revised in 2000, which is an ethical standard for all researchers and all studies including observational studies. The lack of human contact in this study does not exempt the authors from seeking ethical approval. In my opinion, an expedited review could have been sought from the authors’ institution or Health Ministry. Other minor concerns have been raised in the manuscript that can be addressed by the authors.

Reviewer #2: The manuscript is neither technically sound nor add something novel to the literature. Nothing has been done to compenasate for the individual bias that could be introduced during the collection process. Also, no correlation has been made between the number of cases that have been reported at that time in each of the area that has been mentioned.

6. PLOS authors have the option to publish the peer review history of their article (what does this mean?). If published, this will include your full peer review and any attached files.

Reviewer #1: Yes: Sammy Y. Aboagye

Reviewer #2: No

---

## [Author Response · Author response to Decision Letter 0]

7 Aug 2020

RESPONSES TO REVIEWER 1 COMMENTS

Comment-abstract

How many washing places per station should be adequate?

Response: Page page 20, line 380-381

The existing literature on adequacy of and Water, Sanitation and Hygiene facilities (WASH ) facilities in out of home settings (mainly Schools and Health facilities) do not provide standards on adequate number of hand washing stations for transportation stations. We anticipate that if majority of commuters were to follow government hand hygiene recommendations for limiting COVID-19 transmission, having only one washing basin will not be sufficient, given the large number of public transportation users in the city of Accra. Further research will be needed to understand what constitutes adequate washing places per transportation station. 

Please refer to Page page 20, line 380-381

“Studies are needed to determine standards on how many washing places are needed in public places like transport stations.”- 

Studies are needed to determine standards on how many washing places are needed in public places like transport stations.

Comment –abstract-

Were there any other type of hand sanitizer beside alcohol based ones?

Response- Page 14, line 256-258

Our aim was to measure compliance with the preventive measures introduced by the Ghanaian National COVID-19 Response team to limit the spread of the disease. The recommendations specify regular rubbing of hands with alcohol based sanitizers among others. Therefore, we did not capture the availability and use of other non- recommended sanitizers, aside alcohol based sanitizers. Please refer to the data collection tool (Section D & E, questions; 12- 17). 

We have addressed this concern in Page 14, line 256-258

“We did not observe the availability and use of other types of sanitizers, apart from alcohol-based sanitizers which had been recommended by the government” 

Comment

The authors have indicated that no approval was sought for this observational study because 1) there was no direct contact with human subjects, and 2) an anticipated lock-down of the country. The Helsinki Declaration of 1975, revised in 2000, which is an ethical standards for international committee of medical journal editors, stipulates that authors are mandated to indicate whether the procedure employed follows ethical standards by institutional/ national ethical review boards. The lack of human contact in this study does not exempt the study from ethical review since the Helsinki Declaration is applicable to all types of studies including observational ones.

Even though laws regarding the conduct of biomedical research vary country by country, the reasons given by the authors does not absolve them from seeking approval. Ethical review boards both institutional and national, knowing the gravity of Covid-19 pandemic. Instituted expedited reviews for covid-19 related research to allow researchers to share current information on the pandemic. Authors should take advantage and obtained an expedited review or exemption letter from the IRB/Health Ministry.

Response 

The research qualified for exempt from ethics review because: 1) the study was planned for implementation in a public space where there is no expectation of privacy; and 2) the study procedures does not require direct interaction with human subjects and no individual personal identifiers have been reported in the manuscript. However, the authors were unable to submit the protocol to the IRB for a determination of ethics exemption, as required, because most institutions were preparing for a lockdown of the city and the IRB was not receiving ethics applications for review during that period.

Comment

The authors indicated the study had no human interactions, just observations so what specific data sets were obtained that could have legal implications and ethical concerns bearing in mind no ethical approval was even sought for the study? 

It is best practice for authors to deposit their dataset in institutional/public repositories. This is very important because greater number of emails to corresponding authors that hold study datasets mostly do not get response after publication.

Response 

There are no identifying information or behaviors that can be linked to individuals in this dataset. As such the dataset has been made publicly available at Figshare and the digital object identifier (DOI) is 10.6084/m9.figshare.12644897

Comment-10 

Repetition at line 165 to 166 and line 171-172

Response 

The noted repetition at Line has been deleted. Please check page 9, line 181-183 

Comment-11 

Aside wearing nose mask/face shield (uncommon), is that other PPE that passengers could wear?

Response-Page 14, line 268-269

At the time of data collection, the government had not mandated the use of PPE in the country.

“We observed the use of handkerchiefs, headgears, and personal clothing being used as face masks.”

Comment 12 – page 10, line 202 to 203

Table 1. Separate or section such that station type and District/municipality do not align because the table in its current state makes the frequency totals 90 not 45

Response page 11, line 226-227

Marginal totals have been used to separate the station type and District/municipality as recommended.

Comment 13 –page 12, lines 231 to 232

Were the situations where passengers used their personal hand sanitizers other than those provided at the stations?

Response –page 21, line 386- 388

Our study was designed to observe the availability of hygiene facilities at lorry stations and whether these facilities were used by passengers at the station. Thus, data was not collected on the use of personal sanitizers by passengers. 

We have provided further information in the discussion section to address this concern. 

“It is possible that some passengers may have used their own personal alcohol-based sanitizers before boarding or alighting from vehicles. However, the study was not designed to measure this behavior”.- page 21, line 384- 386 

Comment – page 16 and 17, line 297 to 299 

The idea to implement best and routine public safety standards is very laudable but how will your proposed air travel measures be achieved considering infrastructural limitations in these stations?

Response- page 20, line 367-379

Despite these limitations, the government has already put out regulations to implement some airline-industry type travel measures as described on page 20, line 367-379. To be sustainable, these travel measures should be contextualized within the local situation, so that they can be enforced without compromising their effect.

Comment 15 -page 17, line 308 to 310 

What actionable steps do the authors propose/recommend that transportation services adopt to safeguard passengers?

Response- Page 20, line 367-379 

Specific actions that can contribute to addressing the guidelines adherence gaps observed in the current study include enforcing mandatory wearing of face masks including both transport operators and travelers, frequently cleaning surfaces including door knobs, seats, counter tops, and arm rests of vehicles using appropriate sanitizing agents. Enforcing mandatory hand hygiene will only work if the government, at all levels provides financial incentives and technical support for transport operators to install and enforce use of handwashing and hand sanitizing infrastructure. In addition to these, interventions to ensure appropriate physical distancing and implementation of other relevant guidelines when vehicles are in transit are warranted. Presently, use of face masks is mandated by City of Accra regulations. Specific physical distancing on board public transportation was also established early in the pandemic. The main challenge to implementing these recommended actions will be enforcement. Considering the observed poor adherence to the existing guidelines observed in the transport stations, further research is needed to understand the drivers of behavior change related to these guidelines.

Comment-16 –page 17, line 316

Rephrase “especially at the at lorry stations”

Response –page 18, line 337

The phrase has been rephrased. It reads as follows: “especially at the lorry stations”

Comment 17 -page 18, line 325

Replace There with “They” 

Response –page19, line 346 

“There” has been replaced with” they” as requested.

Comment – 18 Page 18, line 341 to 343

Why is the use of face mask in crowded places a health concern? Should it not rather be non-use of face masks?

Response-page 19, line 359-361 

The sentence has been reworded. It reads as follows: “The non-use of face masks in crowded transportation stations, where the practice of social distancing is almost non-existent is a public health concern”

Comment -19 – pages 19-20, line 357 to 367

What interventions have been taken by the authors based on the study findings at the transport stations and operators? Are they aware of the study outcome knowing they are not privy to publication of the findings in scientific journals? Any education on covid-19 related risks and preventive measures on-going?

Response

A research brief was shared with the National COVID-19 response team in early April 2020 to guide decision making, especially in the transportation sector. In addition, members of the research team have been sharing the study findings and recommendations in multiple electronic and print media.

Comment-

The authors, however, failed to outline the needed stringent preventive measures that transportation managers/owners could adopt at the stations to enhance compliance in the discussion.

Response –Page 20 line 367-381

Thanks for your response. Kindly refer to Page 20, line 367-379 for specific actions to address the risks of COVID-19 transmission

RESPONSES TO REVIEWER 2 COMMENTS

Reviewer #2: 

The manuscript is neither technically sound nor add something novel to the literature. Nothing has been done to compensate for the individual bias that could be introduced during the collection process. Also, no correlation has been made between the numbers of cases that have been reported at that time in each of the area that has been mentioned.

Response

There are no comments in the attached PDF file to help the authors to understand how the reviewer’s conclusion was arrived at. We humbly request the reviewer to kindly provide additional information to help the authors understand what the gaps are that makes the manuscript not technically sound. 

To ensure repeatable observations and minimal inter-observer differences a tool that requires simple observations was designed and pretested prior to data collection. Analyses of the pretest results showed that the observations reported by individual observers were reproducible and valid. We have further described these processes in the methods section, page 7, line 144-145, and page 8, line 159-164 

The study was designed to understand how preventive guidelines were being adhered to. Thus, it was not necessary to link station selection to cases that had been reported. In any case, majority of the reported cases were from the GAR but had not been disaggregated by location.

---

## [Decision Letter · Decision Letter 1]

28 Aug 2020

Limiting Spread of COVID-19 in Ghana: Compliance audit of selected transportation stations in the Greater Accra region of Ghana

PONE-D-20-13058R1

Dear Dr. Aryeetey,

We’re pleased to inform you that your manuscript has been judged scientifically suitable for publication and will be formally accepted for publication once it meets all outstanding technical requirements.

Kind regards,

Francesco Di Gennaro

Academic Editor

PLOS ONE

Additional Editor Comments (optional):

Dear Authors,

congratulations.

Now your article can be accept.

I appreciate the interaction beetwen authors and reviewers.

Reviewers' comments:

Reviewer's Responses to Questions

**Comments to the Author**

1. If the authors have adequately addressed your comments raised in a previous round of review and you feel that this manuscript is now acceptable for publication, you may indicate that here to bypass the “Comments to the Author” section, enter your conflict of interest statement in the “Confidential to Editor” section, and submit your "Accept" recommendation.

Reviewer #1: All comments have been addressed

Reviewer #2: All comments have been addressed

2. Is the manuscript technically sound, and do the data support the conclusions?

Reviewer #1: Yes

Reviewer #2: Partly

3. Has the statistical analysis been performed appropriately and rigorously? 

Reviewer #1: Yes

Reviewer #2: N/A

4. Have the authors made all data underlying the findings in their manuscript fully available?

Reviewer #1: Yes

Reviewer #2: Yes

5. Is the manuscript presented in an intelligible fashion and written in standard English?

Reviewer #1: Yes

Reviewer #2: No

6. Review Comments to the Author

Reviewer #1: The authors have addressed the concerns raised during the initial review and made some modifications in the discussions. I thereby recommend the manuscript be accepted for publication.

Reviewer #2: Most of the comments have been satisfactorly addressed by the authors. It could just have been better to build correlation between the number of cases and measure implemented.

7. PLOS authors have the option to publish the peer review history of their article (what does this mean?). If published, this will include your full peer review and any attached files.

Reviewer #1: No

Reviewer #2: No

---

## [Editor Report · Acceptance letter]

1 Sep 2020

PONE-D-20-13058R1 

Limiting Spread of COVID-19 in Ghana: Compliance audit of selected transportation stations in the Greater Accra region of Ghana 

Dear Dr. Aryeetey:

I'm pleased to inform you that your manuscript has been deemed suitable for publication in PLOS ONE. Congratulations! Your manuscript is now with our production department. 

Kind regards, 

on behalf of

Dr. Francesco Di Gennaro 

Academic Editor

PLOS ONE